# RCA: Region Conditioned Adaptation for Visual Abductive Reasoning

## ABSTRACT

Vision foundational models (e.g., CLIP) show strong generalization on various downstream visual perception tasks. However, their ability to reason beyond mere perception is limited, as they are only pre-trained on image-text pairs that hold semantically equivalent meanings. To tackle this, we propose a simple yet effective *Region Conditioned Adaptation* (RCA), a hybrid parameter-efficient fine-tuning method that equips the frozen CLIP with the ability to infer hypotheses from local visual cues. Specifically, the RCA contains two novel modules: regional prompt generator and Adapter$^+$. The prior encodes "local hints" and "global contexts" into visual prompts separately at fine and coarse-grained levels. The latter enhances the vanilla adapters with a newly designed Map Adapter, that directly steers the focus of attention map with trainable query and key projections. Finally, we train the RCA with a new Dual-Contrastive Loss to regress the visual feature simultaneously toward features of literal description (a.k.a. clue text) and plausible hypothesis (abductive inference text). The loss enables CLIP to maintain both perception and reasoning abilities. Experiments on the Sherlock visual abductive reasoning benchmark show that the RCA significantly outstands previous SOTAs, ranking the 1st on the leaderboards (e.g., Human Acc: RCA 31.74 *vs* CPT-CLIP 29.58, higher =better). We also validate the RCA is generalizable to local perception benchmarks like RefCOCO. We would open-source our codes for future research.

## CCS CONCEPTS

• **Computing methodologies → Activity recognition and understanding**.

## KEYWORDS

Visual Reasoning; Region Prompt; Adapter Tuning

**ACM Reference Format:**
Anonymous Author(s). 2024. RCA: Region Conditioned Adaptation for Visual Abductive Reasoning. In *Proceedings of ACM Conference (Conference'17)*. ACM, New York, NY, USA, 10 pages. https://doi.org/10.1145/nnnnnnn.nnnnnnn

## 1 INTRODUCTION

Visual reasoning refers to the ability to understand, interpret, and rationalize predictions derived from visual inputs. This ability is essential for creating AI systems capable of interacting with the

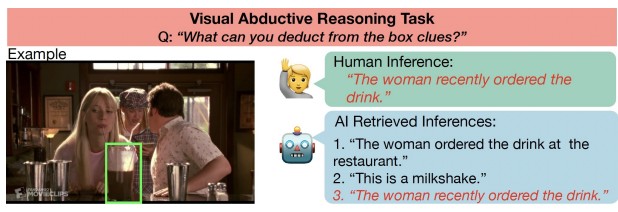

**(a) The Task of Visual Abductive Reasoning**

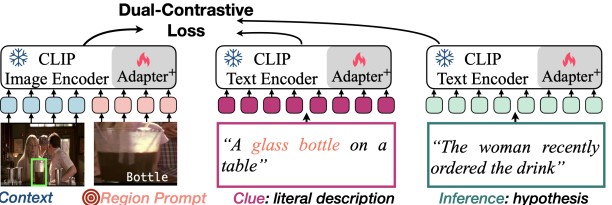

**(b) Region Condtioned Adaptation (RCA)**

Figure 1: Task of Visual Abductive Reasoning (VAR) and Region Conditioned Adaptation (RCA). (a). VAR aims to make the most likely hypotheses from (incomplete) observations; (b). Our region-conditioned adaptation module learns to reason from visual details while keeping CLIP's original ability to align the image with its text description. It allows our model to learn hypotheses that are consistent with the context and learns the causal associations between hypotheses and vision-linguistic observations.

environment [4, 7, 18, 31, 36, 58]. In this regard, abductive reasoning [25] has been a topic of interest in AI for a long time due to its various applications in detecting faults in systems, automated medical diagnosis, and legal reasoning.

Recently, a novel multimodal visual reasoning problem known as Visual Abductive Reasoning (VAR) has been introduced [18], highlighting the significance of integrating both visual and textual modalities to infer logical conclusions from observed image data. Visual abductive reasoning refers to making inferences based on visual information (usually an incomplete set of observations) to arrive at the most plausible (often simplest) explanation or hypothesis for a given observation. For example, in VAR, as shown in Figure 1a, the model is expected to make the inference "*the woman recently ordered the drink*" from the given regional visual hints, which show only the "glass bottle" and surrounding contexts ( "restaurant scene" and "waitress"). Visual abductive reasoning is challenging because it requires a deep understanding of the observed image and the domain (or the context) of the scene depicted in the image. Furthermore, VAR demands the ability to generate hypotheses consistent with the observed visual data and the domain rules. It involves not only recognizing patterns in the images

but also applying domain knowledge, learning to reason about unseen/unknown aspects of the context, and learning the causal relationship between inferences and observations.

Current vision and multimodal foundational models have superior capability in visual understanding and language-based reasoning. However, they are not explicitly modeled to tackle visual abductive reasoning. Interestingly, most of the existing vision-language models are trained in a data-driven manner using image-text contrastive learning [29, 38], image-to-text matching [29] and image-based language generation [29, 33]. However, in visual abduction, there is only a causal association between visual observations and inferences and current models are not trained to tackle this aspect.

Authors in [18] adapted vision foundation models such as CLIP [38] with visual prompts and vision-to-inference based contrastive learning for visual abductive reasoning. Their idea is that if one fine-tunes the CLIP model with vision-to-inference contrastive learning, then the model can learn the domain knowledge explicitly as well as the backward reasoning, i.e., the inference can be made using the observations. However, fine-tuning the entire foundational model is not ideal as that may change the learned representations of the foundational model. Furthermore, direct optimisation of the contrastive loss either using vision-to-inference or vision-to-text-evidence may not allow the model to learn the association between inferences and the observations more effectively.

We also leverage the CLIP model for the VAR task as shown in Figure 1b. However, we resort to parameter-efficient tuning of the CLIP model as a solution. Specifically, we train a few newly added adaptor parameters of vision and text Transformers of the CLIP model using both vision-to-evidence **and** vision-to-inference contrastive losses **jointly**. Our novel adaptor learns new attention maps using low-rank projection matrices, allowing us to learn the semantic associations between the hypothesis and the observations without destroying the semantic knowledge encapsulated in CLIP's vision and text Transformer modules. The optimization of both losses allows us to learn the cause-and-effect relation between the hypothesis (i.e., inference) and observations (i.e., visual and textual evidence). While vision-to-evidence contrastive loss helps to reduce the semantic gap between vision and text modalities using few adaptor parameters, the joint optimization of vision-to-inference and vision-to-evidence contrastive losses helps to learn the causal association between hypothesis and observations (i.e., observations are a result of hypothesis). Furthermore, using newly designed regional prompts, our model attends to the relevant visual cues for hypothesis generation. It helps the CLIP vision Transformer to attend to subtle visual cues without modifying the CLIP vision model and the parameters (–see Figure 1b). These regional prompt tokens are further appended to image context tokens to capture context information. This allows the model to learn context-based domain-level rules and knowledge. For example, during learning our model may learn rules such as "*if it rains, the road can get wet*" or "*in restaurants, there are people, and they order drinks*". This provides the foundational model with relevant visual hints to align textual evidence with vision, associate the hypothesis with multimodal evidence during learning, and learn domain/context-specific knowledge.

Experiments on the Sherlock VAR benchmark show that our model surpasses previous state-of-the-art results, ranking the 1st on the leaderboard[1]. Our contributions are summarised below.

**Region Conditioned Adaptation** (RCA). Our RCA is the first hybrid Parameter-Efficient Fine-Tuning (PEFT) method within the "*prompts + adapter*" paradigm. It guides frozen vision foundation models to make inferences based on visual observations.

**Fine-Grained Region Prompts**. We have designed a new visual prompt that encodes regional hints at a fine-grained level within the CLIP model. Our tests confirm that emphasizing local evidence improves visual abductive reasoning.

**Enhanced Adapter⁺ Tuning**. We present a new Map Adapter that adjusts the attention map using extra query/key projection weights. Our new MAP adapter is orthogonal to the original adapter [47], and they are jointly used to form the Adapter⁺.

**Dual-Contrastive Loss**. We show that joint optimization of vision-to-inference and vision-to-evidence contrastive losses helps to learn the causal association between hypothesis and observations, which aids visual abductive reasoning.

## 2 RELATED WORKS

Our proposed region conditioned adaptation is relevant foundation models, abductive reasoning, parameter-efficient fine-tuning, and fine-grained visual representation learning. We will discuss related works according to the areas below.

**Foundation Models.** Scaling up models' complexities and training data improves the attention-based [14, 16, 35, 37] foundation models' [6, 11, 22, 38, 49, 52] perception capacity, making it proficient in many tasks including zero or few-shot learning. Specifically, Large Language Models (LLM), such as BERT [11], and GPT [6] are trained on large-scale datasets and they generalize to many downstream NLP tasks. Following this trend, several vision-language foundational models are also developed e.g. CLIP [38], ALIGN [22] and BLIP [30]. The main idea behind the vision foundation models is to learn transferable visual representation with noisy text supervision through a two-tower structure. We follow the current baseline of visual abductive reasoning and adopt the CLIP model as the backbone for visual inference.

**Abductive Reasoning Tasks.** Humans make plausible inferences or hypotheses from incomplete observations every day [10]. To teach AI models to attain the same capability, researchers proposed several new tasks, like $\mathcal{ART}$ [3] for NLP, *Sherlock* [18] for vision, and *VideoVAR* [31], *VideoABC* [58] for video. Specifically, the $\mathcal{ART}$ [3] generates the most likely hypothesis (text) to explain what has happened between the two observations (texts). For Sherlock, VideoVAR, and VideoABC, the observations are represented by regional or whole images, while inference is text or middle frames. There are similar tasks, like Visual Commonsense Reasoning (VCR) [53] and Visual7W [64]. Abductive reasoning differs from them in having non-definitely correct, plausible inferences as humans do.

**Parameter-Efficient Fine-Tuning** (PEFT). Transferring foundational models to downstream tasks promotes the development of PEFTs [40]. Representative PEFTs include Prompt, Adapter, and

---

[1]https://leaderboard.allenai.org/sherlock/submissions/public

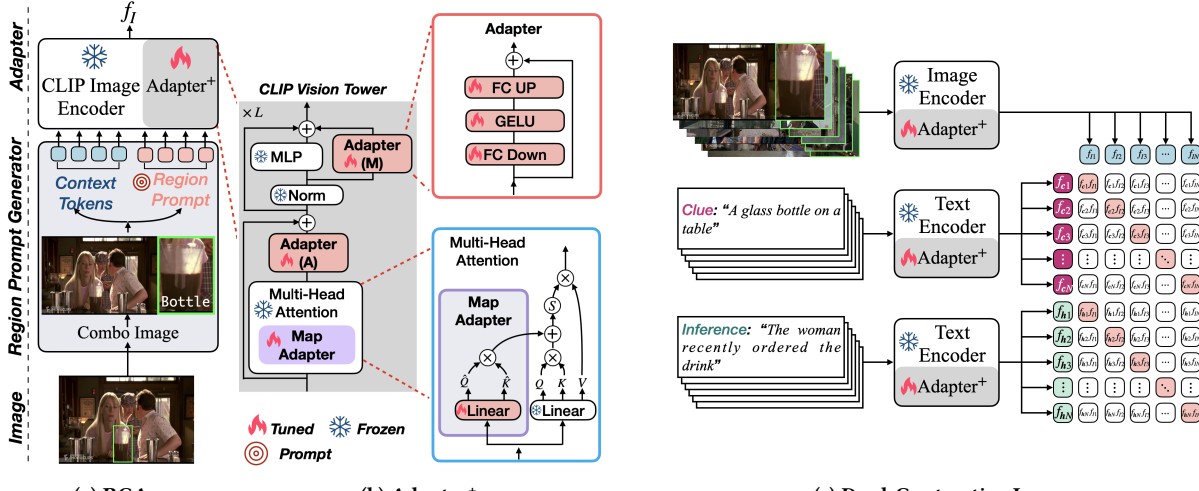

**(a) RCA**  **(b) Adapter$^+$**  **(c) Dual-Contrastive Loss**

**Figure 2: Region Conditioned Adaptation with Dual-Contrastive Loss for Visual Abductive Reasoning. (a)** *RCA* **simultaneously generates region prompt and contextual tokens by intaking a combo-image** $I$**, then tunes the frozen CLIP with Adapter$^+$ on top of reasoning dataset; (b) Adapter$^+$ includes two standard adapters and a novel Map Adapter, which separately adjust token features and the attention map; (c) Dual-Contrastive Loss simultaneously guide the visual content minimize semantic (to clue) and causal (to inference) gaps. (Note: Best viewed in color.)**

LoRA tuning. Specifically, prompt tuning [6, 28, 34, 45, 61] enhances the distinctiveness of inputs by prepending additional tokens, which may be either trainable or fixed. The vision-language prompt tuning can further be divided into textual [12, 13, 62, 63] or visual prompt [2, 23] tuning, depending on the placement of prompt tokens in visual or textual encoders. Certain special visual prompts, such as the Merlot [54], CPT [48], and CiP [42], guide the model to focus on specified areas by overlaying these regions with translucent colours or red circles. In adapter tuning, trainable Multi-Layer Perceptron (mini MLP) [19, 43, 47] or Tiny Attention modules [57] are usually inserted into the foundational model, with only the new additions being fine-tuned. LoRA [20] update parameters using low-rank projections. Our RCA is a hybrid "*prompt+adapter*" tuning to equip the vision foundational models with local reasoning ability, an approach that has not been studied before.

**Fine-Grained Visual Representation.** Our work is also relevant to learning fine-grained visual representation [1, 41, 46, 55, 56, 60] and object detection [5, 17, 21, 24, 59]. Specifically, GLIP [56] and RegionCLIP [60] pre-train foundation models for object detection, supervised by region-text pairs. The former and latter mimic the process of R-CNN[15] and Faster-RCNN [39], generating an object's vector by either encoding the cropped image or RoI pooling. Similarly, UNITER [8] and LXMERT [44] also rely on RoI pooling to generate regional vectors for vanilla vision-language tasks. Besides, the InternImage [46] learns the foundation model with Deformable-CNN for object detection. Other works, such as Object-VLAD [55] for event detection and CLIPTER [1] for scene text recognition, also studied fine-grained modeling. Specifically, the Object-VLAD aggregates densely collected local features with VLAD to generate video representation. The CLIPTER introduces extra cross-attention and the gated fusion module to combine local

and global features. In contrast, our RCA only adjusts the frozen CLIP with an add-on Adapter to tackle new inputs.

## 3 OUR VAR MODEL

### 3.1 Problem Definition

**Problem**: Hessel et al.[18] defines a Visual Abductive Reasoning benchmark named "*Sherlock*" that requires a model to predict the hypothesis from visual observations in "***Observation → Hypothesis***" form. Specifically, visual observation refers to a pre-specified region $r$ of an image $i$ and is accompanied by a clue sentence $c$. Notably, the clue is a straightforward description of real visual content and is only available during training. On the other hand, the hypothesis is defined by an inference sentence $h$. With this, a VAR model calculates a score $s$, which reflects the probability of deducing inference $h$ from the region $r$. Equation (1) shows this scoring function $\mathcal{F}$ and the parameters $\theta$; we call $\mathcal{F}$ the VAR model.

$$s = \mathcal{F}(h, i, r|\theta) \tag{1}$$

A good VAR model should generate a larger matching score when an inference $h$ and observation $\{i, r\}$ are causally related, and a smaller value for wrong or non-related inferences.

### 3.2 Method overview

We introduce the **Region Conditioned Adapter Tuning** (RCA in Figure 2a), which enhances *the vision foundation models to focus on specific visual cues for deducing inference*. The RCA consists of two main parts: a Regional Prompt Generator (RPG in §3.3) for targeting specific visual areas and an Adapter$^+$ module (§3.4) to transfer the frozen CLIP model for reasoning tasks. Finally, we replace Multi-Task Learning [18] with a new **Dual-Contrastive**

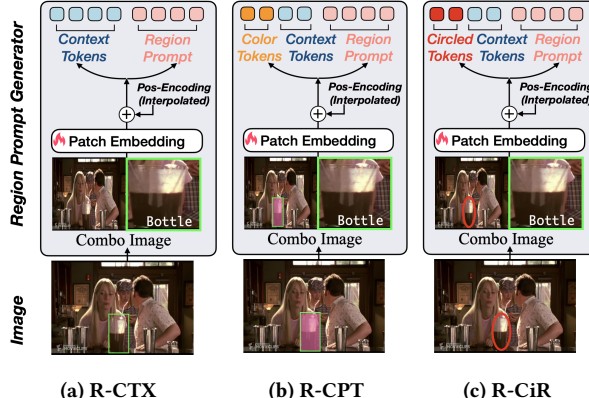

**(a) R-CTX**    **(b) R-CPT**    **(c) R-CiR**

**Figure 3: Three Types of Fine-Grained Region Prompts generated by the RPG. In RPG, we assemble the combo-image I$I$ from region $r$ and context, colorful, or circle-prompted image $i$. (a) R-CTX: *Region+Context*; (b) R-CPT: *Region+Colorful Prompt*; (c) R-CiP: *Region+Circle Prompt*.**

**Loss** (§3.5) to bring the visual features closer to both literal description ("clue") and hypothesis ("Inference"). We will elaborate on each section below.

## 3.3 Regional Prompt Generator

In visual abductive reasoning, it is important to collect all relevant visual cues from the image and the region $r$. Therefore, we use pre-specified observation $r$ as a "prompt" to guide the visual reasoning process directly. Our Regional Prompt Generator (RPG) creates three detailed prompts focusing on specific regions. These prompts harness local features (i.e., region), surrounding context, and existing visual prompts, as shown in Figure 3a-3c. All three types of prompts go through the same process, with the only difference in being whether colors or circles [42] are drawn on the input images. To explain how it works, we'll use the "Region+Context" (R-CTX) as an example.

To prepare prompt and contextual tokens, we pop out patch-embedding layer $\mathcal{F}_{proj}$ and positional encoding PE from the CLIP vision tower $\mathcal{F}_{vis}$. We further resize region $r$ and full image $i$ into squares of the same size, then merge them vertically (or horizontally) into a combined image, i.e., **combo-image I** (eq. (2)). We apply patch-embedding on the combo image and add it to the up-sampled $PE_{inter}$ embedding to get visual tokens $z_0$ (eq. (3)). As the $PE_{inter}$ is twice the size of PE, we initialize it by inflating PE using bilinear interpolation. Notably, $z_0$, generated from the combo-image, already includes both regional prompt and global contextual tokens. The $z_0$ is further fed into the remaining attention and MLP layers (denoted by $\widehat{\mathcal{F}}_{vis}$) to get visual representation $f_I$ (eq. (4)). We unfreeze patch-embedding $\mathcal{F}_{proj}$ and positional encoding $PE_{inter}$ to generate learnable soft prompts.

$$I = Concat\left(\begin{bmatrix} r \\ i \end{bmatrix}\right) \tag{2}$$

$$z_0 = \mathcal{F}_{proj}(I) + PE_{inter} \tag{3}$$

$$f_I = \widehat{\mathcal{F}}_{vis}(z_0) \tag{4}$$

For the "Region + Colorful Prompt" and "Region + Circle Prompt" (R-CPT & R-CiR), we create the combo-image $I'$ from the squared region $r$ and a modified image $i'$. In this image $i'$, we either color the pixels inside the region with a translucent pink rectangle or outline them with a red circle [42]. The rest of the process remains similar to the "Region+Context" prompt.

**Mixed Prompts**: During training, we randomly choose from the three types of prompts (R-CTX, R-CPT, and R-CiR) with equal chance. During testing, we take the average of the visual representations created by these three prompts. This strategy regulates the training process and allows us to obtain better generalizability.

## 3.4 Adapter⁺ Tuning

Adapter tuning adjusts a parameter-frozen foundational model for downstream tasks by fine-tuning a few newly implanted modules (parameters). This strategy is widely used in NLP [47] and computer vision [38] prior works. Current adapters, like mini MLP [47] and tiny attention modules [57], focus mainly on refining visual features. However, they don't consider the need to adjust the original attention maps of the base models. In some inference tasks such as visual abductive reasoning, it is beneficial to adapt the attention maps as well, especially to learn context-based domain knowledge and finer visual details. To tackle this, we augment the vanilla adapter with a new *Map Adapter*, which precisely adapts attention maps in Transformers. This results in the improved Adapter⁺.

The Adapter⁺ pipeline is illustrated in Figure 2b. We first include two basic adapters, referred to as Adapter (A&M). They are placed after the MSHA module and parallel to the MLP module in the $l$-th encoder of a CLIP tower (e.g., $\mathcal{F}_{vis}$ or $\mathcal{F}_{txt}$). These adapters are shallow and contain only two fully-connected layers to downgrade and upgrade feature dimension ($\mathbb{R}^D \leftrightarrows \mathbb{R}^d, d < D$) with a GELU activation in between (Equation 5-7). The light red font indicates the parameters in the modules are tuned.

$$z'_l = \text{MHSA}(z_{l-1}), \ l = 1, 2, \cdots, L \tag{5}$$

$$z''_l = \text{Adapter\_A}(z'_l) + z_{l-1}, \tag{6}$$

$$z_l = \text{MLP}(z''_l) + \text{Adapter\_M}(z''_l) + z''_l, \tag{7}$$

The **Map Adapter** further refines the MSHA module by adding a small, modified attention map, labeled as $\widehat{Q}\widehat{K}^T$ (refer to eq. (8)). This additive map helps to adjust the original attention map dynamically, improving the model's ability to focus on relevant information. To ensure that the original attention map isn't altered too much, we use simpler $D \to d$ projections for generating the query and key (see eq. (9)). Here, $d$ is smaller than $D$.

$$z'_l = \text{Softmax}\left(\frac{QK^T + \widehat{Q}\widehat{K}^T}{\sqrt{D}}\right)V, \tag{8}$$

$$\widehat{Q}, \widehat{K} = z_{l-1} \times \widehat{W}_q, \widehat{W}_k, \ \widehat{W}_{q,k} \in \mathbb{R}^{D \times d} \tag{9}$$

$$Q, K, V = z_{l-1} \times W_q, W_k, W_v, W_{q,k,v} \in \mathbb{R}^{D \times D} \tag{10}$$

We compared the enhanced Adapter⁺ with min MLP and Tiny Adapter counterparts (see §4.4) and found that our tuning method consistently performs better.

## 3.5 Dual-Contrastive Loss

As an observation contains three modalities, such as visual $I$, clue sentence $c$, and inference sentence $h$, we comprehensively study their mutual influences by deploying contrastive loss between different modalities pairs. Specifically, as shown in Fig. 4, we deploy dual, triple, and single contrastive loss in the training phase and screen out that the *Dual-Contrastive Loss works best* (Fig. 2c). We first elaborate on the Dual-Contrastive Loss and then compare it with the other counterparts.

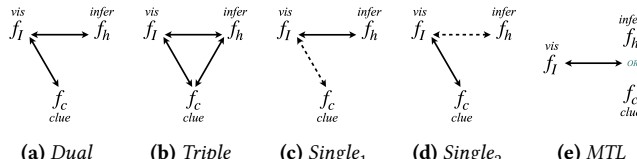

**(a)** *Dual*  **(b)** *Triple*  **(c)** *Single₁*  **(d)** *Single₂*  **(e)** *MTL*

**Figure 4: Training Losses: Dual (a), Triple (b) and Single (c)-(e) contrastive loss for learning to match visual $I$ to texts $c$ and $h$. Solid and dashed lines represent the presence or absence of contrastive loss during training.**

**Dual-Contrastive Loss**: Both the clue $c$ and inference $h$ are positively relevant to visual $I$. More specifically, the former is literally equivalent, while the latter is causally related to the visual hints. Although their relations are in different forms, we can still deploy a Dual-Contrastive Loss, including one for "*vision-clue*" pair and the other for "*vision-inference*" pair (Fig. 4a), to regress visual features toward two textual targets. We use CLIP text tower $\mathcal{F}_{txt}$ to extract features for clue $c$ and inference $h$. We use Equation (4) to extract visual feature $f_I$ for the observation $I$. The mathematical process is present in Equations (11)-(12).

$$loss_{dual} = \mathcal{L}_{contrast}(f_I, f_c) + \mathcal{L}_{contrast}(f_I, f_h) \quad (11)$$

$$f_c = \mathcal{F}_{txt}(c), \quad f_h = \mathcal{F}_{txt}(h) \quad (12)$$

**Other Loss Variants**. The rest loss functions include the **Triple** and **Single** contrastive loss. Particularly, compared with dual contrastive loss, the triple one newly adds the "*inference-clue*" pair (e.g., Fig. 4b and Eq. 13).

$$loss_{triple} = \mathcal{L}_{contrast}(f_I, f_c) + \mathcal{L}_{contrast}(f_I, f_h) + \mathcal{L}_{contrast}(f_c, f_h) \quad (13)$$

We later observed that: additional *inference-clue* loss in triple contrastive hurts overall performance, as the two texts (i.e., clue and inference) are not literally equivalent. For example, the clue sentence "*the road is wet*" $\neq$ inference sentence "it has rained before". Therefore, we can only let the two texts learn toward the same third-party feature (e.g., the visual) instead of artificially forcing them to be equivalent.

For the single contrastive loss, we have three options, namely *vision-inference* (Fig. 4c), *vision-clue* (Fig. 4d), *multi-task* learning (MTL in Fig.4e). Notably, we use an identical textual encoder for clue and inference during testing, since we only learn a single contrastive loss between a pair of modalities during training.

These three options can be expressed in one unified form (Eq. 14), by thresholding a random probability $p$ with different values $\overline{T}$. Specifically, when $\overline{T}$ = 1.0 or 0 or 0.5, the single contrastive loss become the *vision-clue*, *vision-inference* and *multi-task* learning loss [18].

$$loss_{single} = \mathcal{L}_{contrast}(f_I, f_{txt}), \quad f_{txt} = \begin{cases} \mathcal{F}_{txt}(h), p > \overline{T} \\ \mathcal{F}_{txt}(c), p < \overline{T} \end{cases} \quad (14)$$

With the single contrastive loss, we find that only minimizing the gap between a pair, such as *vision-clue* (or *vision-inference*) will also shorten the gap between the other pair *vision-inference* (or *vision-clue*), indicating retrieval and abductive reasoning tasks are positively correlated. We give detailed analysis in §4.4.

## 4 EXPERIMENTS

We comprehensively study the RCA and Dual-Contrastive Loss on the Sherlock benchmark [18]. We also tested the RCA's adaptability on the RefCOCO [51], which focuses on grounding expression to regions. We present details below.

### 4.1 Datasets

The **Sherlock** dataset [18] contains 103K images collected from the Visual Genome [27] and Visual Common Sense Reasoning [53] datasets. These images are split into 90K training, 6.6K validation, and 6.6K testing sets. Each image is re-annotated with an average of 3.5 observation-inference pairs, forming 363K samples. Particularly, a sample includes a bounding box $r$ and two texts (i.e., clue $c$ + inference $h$). Notably, the validation set can be evaluated offline with officially provided scripts, while the testing set needs to be submitted to the evaluation server of leaderboard.

Three types of evaluation metrics, from *retrieval*, *localization*, and *comparision* aspects, are adopted for this benchmark. Specifically, retrieval metrics include $img \leftrightarrows text$ mean rank, $P@1_{i\to t}$. For localization, accuracies of grounding candidate regions to the inferences are adopted. Comparison metric calculates the accordance between machine and human predictions.

The **RefCOCO** dataset [51] origins from the MSCOCO dataset [32]. We test the generalization of the RCA on the Referring Expression Comprehension (REC task) using Accuracy@0.5. This task aims to link a distinctive sentence to a specific object box when multiple similar objects are present. This dataset contains 3 splits: RefCOCO, RefCOCO+, and RefCOCOg. The RefCOCO and RefCOCOg allow for relational expressions of position (left/right), while RefCOCO+ has only expression on appearance. Specifically, RefCOCO/+/g contains 19.9/19.9/26.7K images, respectively, covering 50.0/49.8/54.8K object instances with corresponding 142/141/85K referring expressions. Since REC requires bounding box proposals for the "*text-to-region*" grounding, we adopt the YoloV8 to generate candidate proposals as inputs for our RCA.

### 4.2 Implementations

We implement the RCA and Dual Contrastive Loss on top of the OpenCLIP [9, 38] PyTorch toolkit[2], and fix the training & testing recipe to be the same for all ablations unless otherwise stated.

**Training**. We resize $r$ and $i$ into 224×224 (336 for high resolution) square images and then concatenate them into combo-image $I$ of size 448×224. We initialize CLIP from OpenAI pre-trained weight

---

[2]https://github.com/mlfoundations/open_clip

Table 1: Comparison with state-of-the-art methods using the Sherlock Testing Leaderboard.
The up arrow ↑ (or down arrow ↓) indicates the higher (or lower), the better.

| Test-Set | | Parameters | Retrieval | | | Localization | Comparison |
|---|---|---|---|---|---|---|---|
| Model | Backbone | Tuned (M↓) | im→txt (↓) | txt→im (↓) | P@1$_{i→t}$ (↑) | GT/Auto-Box (↑) | Human Acc (↑) |
| LXMERT [44] from [18] | F–RCNN | NA | 51.10 | 48.80 | 14.90 | 69.50 / 30.30 | 21.10 |
| UNITER [8] from [18] | | NA | 40.40 | 40.00 | 19.80 | 73.00 / 33.30 | 22.90 |
| CPT [48] from [18] | RN50×64 | NA | 16.35 | 17.72 | 33.44 | 87.22 / 40.60 | 27.12 |
| CPT [48] from [18] | | 149.62 | 19.85 | 21.64 | 30.56 | 85.33 / 38.60 | 21.31 |
| CPT [48] (our impl) | | 149.77 | 19.46 | 21.14 | 31.19 | 85.00 / 38.84 | 23.09 |
| Full Fine-Tuning (R–CTX) | | 149.77 | 15.63 | 18.20 | 33.76 | 86.19 / 40.78 | 27.32 |
| Our RCA (R–CTX) | ViT–B–16 | 42.26 | 15.59 | 18.04 | 33.83 | 86.36 / 40.79 | 26.39 |
| ↳ Mixed Prompts | | 42.26 | 14.39 | 16.91 | 34.84 | 87.73 / 41.64 | 26.11 |
| ↳ Dual-Contrast Loss | | 42.26 | **13.92** | **16.58** | **35.42** | **88.08** / **42.32** | **27.51** |
| CPT [48] (our impl) | | 428.53 | 13.08 | 14.91 | 37.21 | 87.85 / 41.99 | 29.58 |
| Our RCA (R–CTX) | ViT–L–14 | 89.63 | 11.36 | 13.87 | 38.55 | 88.68 / 42.30 | 31.72 |
| ↳ Mixed Prompts | (336) | 89.63 | 10.48 | 12.95 | 39.68 | 89.66 / 43.61 | 31.23 |
| ↳ Dual-Contrast Loss | | 89.63 | **10.14** | **12.65** | **40.36** | **89.72** / **44.73** | **31.74** |

and tuning for 10 epochs with a cosine learning lr schedule. We train with a global batch size=3200, lr=2e-4 using ViT-B-16 backbone (batch=400, lr=2e-5 for ViT-L-14-336) on 2×80 GB A100 GPUs.

**Testing**. We apply the same preprocess for region $r$ and full image $i$ to prepare combo-image $I$ as the training phase. Given a set of visuals $\{r, i\} \times K$ and inferences $\{h\} \times K$, we first calculate the $K \times K$ matrix of *vision-inference* similarity and report retrieval, localization and comparison metrics based on the matrix.

## 4.3 Comparision with the State-of-the-Art

We compare RCA with the SOTAs on the Sherlock test set. These results are evaluated and published on the official leaderboards.

As shown in Table 1, our **RCA ranks the 1st** on the Sherlock Leaderboard regarding most of the evaluation metrics. It significantly outperforms SOTA competitors. For example, our model achieves a "Human Acc" score of **31.74**, compared to 29.58, 22.90, and 21.10 for CPT-CLIP, UNITER, and LXMERT models. We note that models built on the CLIP model, including ours and CPT-CLIP, perform much better than traditional models like UNITER and LXMERT. This suggests that large-scale pre-trained knowledge is beneficial for tasks requiring abductive reasoning. We further validate that our RCA performs well with fine-grained regional evidence as a prompt for visual reasoning tasks. Our model achieves a Human Acc score of 26.39/31.74 (↑3.30/2.16), compared to 23.09/29.58 for CPT-CLIP when using different backbones. Lastly, our new "Dual-Contrastive Loss" feature further enhances the performance of the RCA. In summary, our model with Dual-Contrastive Loss outperforms current state-of-the-art methods.

## 4.4 Ablation Study

This section comprehensively studies various factors that influence the performance of RCA on the validation set. We use Mixed Prompts, Dual-Contrastive Loss and ViT-B-16 as default settings, except in the comparison of different prompts and losses. More ablations are in supplementary.

**Impacts of Integrating Adapters**. We analyze how our model performs when we remove certain components, specifically the vanilla (A&M) and Map Adapters, one at a time. The results in Table 2 show that performance decreases with fewer adapters. Specifically, using all three types of adapters produces the best results under most evaluation metrics. "Adapter (M)" is the best choice when limited to using just one type of adapter. If we can use two types, the best combination is "Adapter (M) + Map Adapter", suggesting that the Map Adapter complements the vanilla adapter well.

**Effects of Fine-Grained Regional Prompts**. We also explore how adding fine-grained regional prompts influences the performance of existing prompting techniques, such as colorful (CPT in [48]) and circle prompts (CiP in [42]). In Table 3, the terms "Region Only" and "Context" refer to feeding either just the regional box part or the entire image into the CLIP vision tower, respectively.

We observe that adding fine-grained tokens based on regional cues significantly improves the performance of all coarse-grained prompts, including "Context", "CPT", and "CiP" across all metrics. This basically verifies that "global context + local cues" complement each other well for abductive reasoning. Moreover, we test the Mixed Prompt mode described in §3.3 and observe a stable performance for most metrics.

**Dual-Contrastive Loss** *vs* **Single/Triple counterparts**. We test different types of contrastive losses using our RCA model. In Table 4, the Dual-Contrastive loss performs better than the Multi-Task Learning and the other single & triple counterparts under most metrics. In terms of localization, the Dual-Contrastive loss is slightly lower than its MTL counterparts but still shows a very competing performance.

We further look into the individual contrastive loss value between modality pairs on the validation set to understand how modalities mutually influence each other. Specifically, we first report the loss value between each pair before the training phase (i.e., No Train or Zero-Shot reasoning), then re-calculate them after the model is trained with different losses (Fig 5).

We observe that the gaps of *vision-clue* and *vision-inference* are positively correlated. Specifically, when we minimize one of the

**Table 2: Impacts of integrating adapters. We compare the single, dual, and triple adapters.**

| Val-Set | Adapter Types | | | Parameters | Retrieval | | | Localization | Comparison |
|---|---|---|---|---|---|---|---|---|---|
| Adapters | Adapter_M | Adapter_A | Map Adapter | Tuned (M↓) | im→txt (↓) | txt→im (↓) | P@1$_{i→t}$ (↑) | GT/Auto-Box (↑) | Human Acc (↑) |
| ↳×1 | | | ✓ | 32.00 | 19.35 | 22.56 | 30.46 | 85.77 / 39.27 | 22.89 |
| ↳×1 | | ✓ | | 32.01 | 18.52 | 21.53 | 31.40 | 85.85 / 39.29 | 23.63 |
| ↳×1 | ✓ | | | 32.01 | 14.79 | 17.14 | 34.82 | 87.82 / **42.32** | 26.66 |
| ↳×2 | | ✓ | ✓ | 37.13 | 16.41 | 18.98 | 33.11 | 86.96 / 40.28 | 24.99 |
| ↳×2 | ✓ | ✓ | | 37.14 | 14.75 | 16.76 | 35.14 | 87.89 / 41.08 | 26.78 |
| ↳×2 | ✓ | | ✓ | 37.13 | 14.51 | 16.54 | 35.15 | 88.06 / 41.34 | 26.41 |
| RCA ↳×3 | ✓ | ✓ | ✓ | 42.26 | **14.26** | **16.44** | **35.46** | **88.23** / 41.91 | **26.80** |

**Table 3: Impacts of Fine-Grained Regional Prompts.**

| Val-Set | Retrieval | | | Localization | Comparison |
|---|---|---|---|---|---|
| Prompt Type | im→txt (↓) | txt→im (↓) | P@1$_{i→t}$ (↑) | GT/Auto-Box (↑) | Human Acc (↑) |
| Region Only | 22.76 | 22.62 | 30.44 | 86.10 / 41.86 | 23.66 |
| Context | 45.28 | 54.57 | 18.12 | NA | 21.99 |
| ↳ + Region (R-CTX) | 15.24 (-30.04) | 17.22 (-37.35) | 34.29 (+15.88) | 87.23 / 41.11 | 26.69 (+4.70) |
| CPT [48] | 17.99 | 19.71 | 31.94 | 86.22 / 39.98 | 25.53 |
| ↳ + Region (R-CPT) | 14.30 (-3.69) | **16.17** (-3.54) | 35.57 (+3.63) | 87.91 / 42.18 (+1.69 / 2.22) | 26.21 (+0.68) |
| CiP [42] | 18.08 | 19.89 | 31.71 | 85.85 / 39.99 | 24.11 |
| ↳ + Region (R-CiP) | 14.27 (-3.81) | 16.25 (-3.64) | **35.61** (+3.90) | 87.90 / **42.62** (+2.05 / 2.63) | 26.51 (+2.40) |
| Mixed Prompts (RPA) | **14.26** | 16.44 | 35.46 | **88.23** / 41.91 | **26.80** |

**Table 4: Comparison of Different Losses on Sherlock Val Set.**

| Val-Set | Retrieval | Localization | Comparison |
|---|---|---|---|
| Loss Type | P@1$_{i→t}$ (↑) | GT/Auto-Box (↑) | Human Acc (↑) |
| Single$_2$ Loss | 25.29 | 82.52 / 30.23 | 21.64 |
| Single$_1$ Loss | 34.57 | 87.96 / 41.60 | 25.64 |
| MTL [18] | 34.82 | 87.63 / **42.33** | 26.07 |
| Triple-Loss | 35.40 | 88.22 / 41.91 | 25.31 |
| **Dual-Loss** | **35.46** | **88.23** / 41.91 | **26.80** |

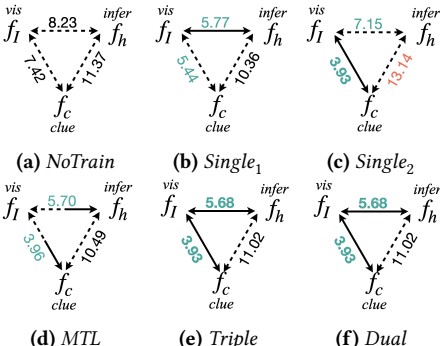

**(a)** *NoTrain*  **(b)** *Single$_1$*  **(c)** *Single$_2$*

**(d)** *MTL*  **(e)** *Triple*  **(f)** *Dual*

**Figure 5: Contrastive Losses between Modality Pair on Sherlock Validation Set. We used a model trained with (a) No Training, (b-c) Single, (d) MTL, (e) Triple, and Dual (f) losses. The Green/Red implies the decreasing/increasing of loss values, compared with No Training. Solid and dashed lines denotes the presence or absence of contrastive loss during training.**

gaps in training, the other one will also become smaller (e.g., Fig.

5b-5c). Whereas, the gap of *inference-clue* seems not to correlate to gaps of *vision-clue* and *-inference*, as the former is slightly closer or even larger after minimizing either of the latter gaps (e.g., red/black value in Fig 5b-5d, and 5f). If we enforce the model to close the *inference-clue* gap during training, the *vision-clue* and *-inference* gap would become larger (Triple *vs* Dual, Fig. 5e *vs* 5f). The reason is that the clue and inference sentences are not literally equivalent and better to be bridged by an extra rational process.

**Influence of Bottleneck Dimension** $d$ **in Adapters.** We study the influence of different bottleneck dimensions in the RCA, ranging in $d = \{\frac{D}{32}, \frac{D}{16}, \frac{D}{8}, \frac{D}{4}, \frac{D}{2}, D\}$. Notably, a higher $d$ basically introduced more tuned parameters and larger FLOPs, as shown in Figure 6a. For the retrieval metrics, such as mean $img \leftrightarrows txt$ rank, a lower value indicates better performance, whereas the rest are the opposite. We observe from Fig. 6a-6d that an optimal choice is $d = \frac{D}{4}$, indicating that adjusts the frozen foundational model with either a very heavy $D$ or lightweight $\frac{D}{32}$ would result in suboptimal performance. Notably, human accordance is influenced by the human's subjective judgement and has a different trend. Overall, we fix $d = \frac{D}{4}$ for all following experiments.

**Influence of Adapting CLIP Vision/Text Tower.** The CLIP follows a two-tower design, each tower separately for visual/textual embedding; thereby, we can independently insert adapters into visual and textual towers to assess their contributions. We test inserting Adapter$^+$ into the "*Only Text*" tower, "*Only Vision*" tower, and both towers. As shown in Table 5, adapting both CLIP vision and text towers performs the best at the cost of the most tuned parameters among the three options. Notably, both "*Only Vision*" and "*Only Text*" has a large margin in performance compared with the "*Vision + Text*", indicating the adaptions on two towers are complementary.

**Figure 6: Performance of different $d$ in adapters. Best viewed in color.**

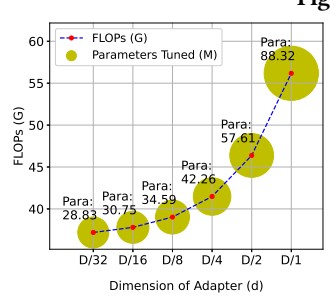

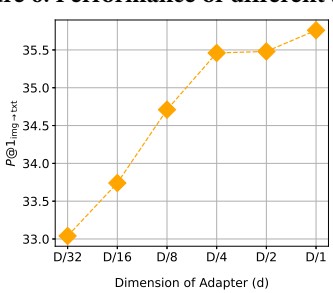

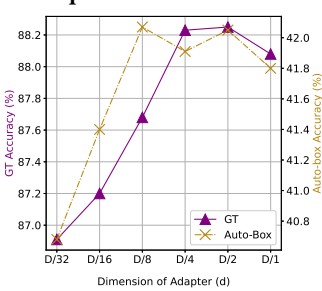

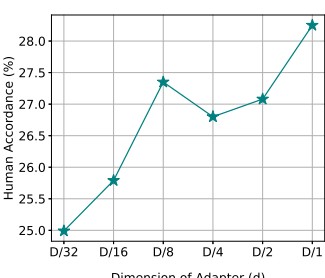

**(a) FLOPs and Tuned Params (↓)**     **(b) Retrieval P@1$_{img→txt}$ (↑)**     **(c) Localization (↑)**     **(d) Human Accordance (↑)**

**Table 5: Influence of Adapting CLIP Vision/Text Tower.**

| Val-Set Towers | Parameters Tuned (M↓) | Retrieval P@1$_{i→t}$ (↑) | Localization GT/Auto-Box (↑) | Comparison Human Acc (↑) |
|---|---|---|---|---|
| Only Text | **31.62** | 25.72 | 77.71 / 34.08 | 20.67 |
| Only Vision | 37.53 | 31.79 | 86.93 / 40.38 | 24.02 |
| Vision+Text | 42.26 | **35.46** | **88.23 / 41.91** | **26.80** |

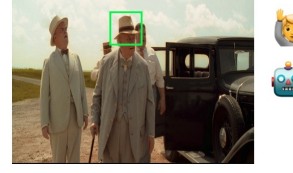

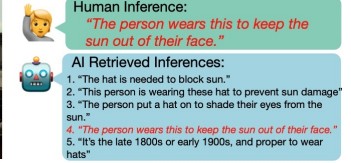

**(a) Visualization 1**

**(b) Visualization 2**

**Figure 7: Qualitative results obtained by rpa. The machine retrieves the top-5 most likely inferences according to the box region. Red sentence indicates that the machine finds the same inference as a human expert.**

## 5 QUALITATIVE RESULTS OF RCA

We present two qualitative examples obtained by the RCA in Figure 7 and more examples in supplementary. Specifically, a human expert gives a possible inference from given a regional cue specified by the box, and the machine retrieves the top-5 most likely inferences. The sentence indicates the correct match with the human's performance. We observe from Example 1 & 2 that the machine manages to deduct human-like inference such as "*He is trying to on board*" from an observation of "*a man under an airplane*" and "*prevent the sunshine*" from "*a wearing hat*".

## 6 GENERALIZATION ON REFCOCO

We also tested the generalization of the RCA on the RefCOCO dataset using a two-stage pipeline in Table 6. Specifically, we employed YoloV8 as the object detector to propose candidate object

boxes. Then, we utilized the RCA to align textual sentences with the object box with the highest matching score. We evaluated the RCA using a single-prompt mode, such as "R-CTX", "R-CPT", to observe their respective effects.

**Table 6: Comparsion on the RefCOCO+/g. We test with RCA ViT-L14 (336). higher=better**

| Model | RefCOCO (↑) | | | RefCOCO+ (↑) | | | RefCOCOg (↑) | |
|---|---|---|---|---|---|---|---|---|
| | val | testA | testB | val | testA | testB | val | test |
| MAttNet [50] | 76.65 | 81.14 | 69.99 | 65.33 | 71.62 | 56.02 | 66.58 | 67.27 |
| UNITER$_L$ [8] | 81.41 | 87.04 | 74.17 | 75.90 | 81.45 | 66.70 | 74.86 | 75.77 |
| MDETR [26] | **86.75** | **89.58** | **81.41** | **79.52** | 84.09 | **70.62** | **81.64** | **80.89** |
| RCA (R−CTX) | 73.31 | 81.95 | 60.84 | 75.81 | **86.47** | 61.98 | 76.35 | 75.43 |
| RCA (R−CPT) | 74.04 | 82.80 | 62.81 | 76.40 | 86.34 | 63.12 | 76.59 | 75.47 |

The RCA performs better than two-stage models like MAttNet on the RefCOCO+/g sets, emphasizing appearance descriptions (e.g., "*a person with a yellow tie*"). However, it lags behind MAttNet on RefCOCO, which focuses on positional descriptions (e.g., "*left person*"). This discrepancy arises because MAttNet explicitly encodes *appearance*, *location*, and *relation* information, while RCA only encodes appearance. Although RCA is adaptable to the Referring Comprehension task, it falls behind one-stage end-to-end models like MDETR. The advantage of MDETR has comes from its design to simultaneously regress box coordinates and establish visual-linguistic alignment, especially for visual grounding. In contrast, our RCA have to rely on third-party proposals from YoloV8.

## 7 CONCLUSION

We propose a new Region Conditioned Adaptation (RCA) with a Dual-Contrastive Loss for Visual Abductive Reasoning. Specifically, our method validates that curating fine-grained regional prompts is feasible for CLIP tuning, getting back local details, and benefiting abductive reasoning. We also reveal the positive relationships between the VAR and Vanilla Visual Retrieval tasks, unifying their training processes with the Dual-Contrastive Loss. Extensive experiments show that the RCA and the new loss are robust and effective for abductive reasoning and surpass previous SOTAs. The success of the two factors also paves future ways for exploring Multi-Grained, Chain-of-Thoughts Prompts, Visual Referring Prompt, and other multiple relationships modeling on the VAR.

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
