# OpenReview forum: "RCA: Region Conditioned Adaptation for Visual Abductive Reasoning"
_acmmm.org/ACMMM/2024/Conference — MM2024 Poster_

### Official Review · Reviewer_E3XH · 2024-05-24

**Rating:** 3
**Confidence:** 3

**Summary:**

To tackle the Visual Abductive Reasoning (VAR) task, this paper propose a Region Conditioned Adaptation (RCA) framework with
a Dual-Contrastive Loss. In their design, the vision foundational model, CLIP, is leveraged to tackle the VAR task through an eco-friendly parameter-efficient fine-tuning design. The RCA primarily features with two modules. A regional prompt generator leverages three different strategies to emphasize the regional prompts for enhanced abductive reasoning performance. And an Adapter+. introduces a Map Adapter to adjust the attention maps for feature encoding. The Dual-Contrastive loss jointly optimize the vision-to-inference and vision-to-evidence contrastive losses for a better visual abductive reasoning performance. Extensive experiments are carried out on two different benchmark datasets to validate the effectiveness of the proposed method.

**Strengths:**

The authors try to apply the pre-trained large vision-language foundation model CLIP in tackling specific vision-language tasks with as few training parameters as possible.

**Limitations:**

1. The contribution of this work is limited as 1) The authors select and combine two popular Parameter-Efficient Fine-Tuning method by “prompt + adapter”. 2) The proposed Regional Prompt Generator has limited novelty, as this design is a combination of a conventional operation and two designs that have been proven to effectively emphasize local evidence. 3) The Enhanced Adapter+ Tuning is not well explained as the difference between other adapter is inconclusive in lines 425-427. What's the meaning of "consider the need to adjust the original attention maps of the base models"?
2. Table 2 should provide the performance of the base model, i.e., frozen CLIP, to verify the effectiveness of different adapters.
3. In Table 1, all the comparisons come from [18] and the compared methods are limited.
4. The results in Table 6 is confusing. The proposed method is not competitive compared with other methods, and hence the effectiveness of the proposed method is not validated on this dataset. It is confusing why authors select this benchmark that their method is not competitive.
5. References need to be improved, including but not limited to 1) standardizing the format; 2) reducing citations to arXiv. papers (Some papers actually have been accepted but authors still use arxiv format).
6. The writing and format of the manuscript need to be enhanced.

**Suitability:**

2

---

### Official Review · Reviewer_DbjW · 2024-05-26

**Rating:** 3
**Confidence:** 2

**Summary:**

This paper introduces Region Conditioned Adaptation (RCA) to enhance the abductive reasoning ability of vision foundational models. The authors propose two modules: regional prompt generator and Adapter+. The former encodes local hints and global contexts into visual prompts, while the latter augments the vanilla adapter into a Map Adapter with an adjusted attention map. Experimental results on the Sherlock visual abductive reasoning benchmark demonstrate that RCA achieves superior performance compared to previous methods.

**Strengths:**

The paper is well-written and clearly organized. The ablation studies are comprehensive. The proposed method significantly outstands previous SOTAs on the Sherlock abductive reasoning benchmark.

**Limitations:**

1. The authors mainly focus on visual abductive reasoning, but the experimental validation is insufficient. As far as I know, besides Sherlock, there are commonly used datasets including VAR[1] and DHPR[2]. The authors should validate the proposed methods on multiple datasets.
2. In Table 1, the authors compare their approach with traditional models; however, there might be an unfair comparison due to the differing backbones used in these methods.
3. The model name in the caption of Figure 7 appears to be incorrect.
4. Could you clarify the role of the "c" in line 325? Is there a missing representation of "c" in Equation (1)?

[1]Liang C et al. Visual abductive reasoning.CVPR, 2022.
[2]Charoenpitaks K et al. Visual Abductive Reasoning Meets Driving Hazard Prediction: Problem Formulation and Dataset. 2023.

**Suitability:**

2

---

### Official Review · Reviewer_ynC6 · 2024-05-27

**Rating:** 4
**Confidence:** 4

**Summary:**

This paper proposes a hybrid parameter-efficient fine-tuning method that equips the frozen CLIP with the ability to infer hypotheses from local visual cues. It contains a regional prompt generator to encode “local hints” and “global contexts” into visual prompts separately at fine and coarse-grained levels and an Adapter module to enhance the vanilla adapters with a newly designed Map Adapter, that directly steers the focus of attention map with trainable query and key projections. In addition, they propose a DualContrastive Loss to regress the visual feature simultaneously toward features of literal description and plausible hypothesis. Experiments on the Sherlock visual abductive reasoning benchmark show that their method significantly outstands previous SOTAs. Ablation studies are also presented, showing the significance of each proposed technique.

**Strengths:**

1. I think this task is interesting.
2. This paper proposes a hybrid Parameter-Efficient Fine-Tuning (PEFT) method within the“prompts + adapter” paradigm.
3. Authors propose a new Map Adapter that adjusts the attention map using extra query/key projection weights.
4. This paper is well written.

**Limitations:**

1. In Table 1, I find that the CPT results for vit-large are already better than the RCA results for vit-b, does this mean that the model just can not see specific details due to the low resolution, instead of lacking the ability to infer?
2. In Table 2, you can add your baseline, is it Full Fine-Tuning (R-CTX) in Table 1?
3. I think the results of Table 2 can be further analyzed. In the table, Adapter_M has the best result, the Adapter_A and Map Adapter are next in order. Is this linearly related to the number of parameters of the adapter? In addition, in conjunction with Adapter_M, the Map Adapter performs a little better, why is that?
4. In my opinion, the Adapter is similar to a LoRA module, Have any experiments been done to use LoRA directly?
5. In lines 507-512, authors talk about clues and inferences that may not be equivalent. But the clues match with the images, and if the inferences and the clues don't match, the inferences do not match with the images.  I think maybe the explanation is not very intuitive.

**Suitability:**

3

---

### Official Review · Reviewer_c4TM · 2024-05-27

**Rating:** 3
**Confidence:** 3

**Summary:**

The authors propose a simple yet effective Region Conditioned Adaptation (RCA), a hybrid parameter-efficient fine-tuning method that equips the frozen CLIP with the ability to infer hypotheses from local visual cues. Specifically, RCA includes two novel modules: a regional prompt generator and Adapter+. The former encodes "local hints" and "global contexts" into visual prompts at fine and coarse-grained levels, respectively. The latter enhances the standard adapters with a newly designed Map Adapter, which directly guides the focus of the attention map using trainable query and key projections. Finally, we train the RCA with a new Dual-Contrastive Loss to regress the visual features simultaneously towards the features of literal descriptions (i.e., clue text) and plausible hypotheses (abductive inference text).

**Strengths:**

1. Visual Abductive Reasoning (VAR) is a very interesting research problem, and the RCA proposed by the authors shows some innovation.

2. The training strategy in RPG is reasonable. Especially, mixing R-CTX, R-CPT, and R-CiR prompts is reasonable.

3. The ablation experiments are designed comprehensively. Some training details are provided, and the authors promise to open-source the code.

**Limitations:**

1. In the introduction, the motivation for incorporating local visual reasoning hasn't been adequately explained. The statement "as they are only pre-trained on image-text pairs that hold semantically equivalent meanings" is too general.

2. The explanation of how the Map Adapter adjusts the attention map in Adapter+ was not clear. I believe you need a logically sound improvement rationale to drive this contribution.

3. Why did you choose to discard the direct connection between clue and inference when designing the Dual-Contrastive Loss, instead of adding a regularization loss? I believe this aspect of the design requires further discussion.

4. Some of the SOTA models are from arXiv papers. Are these commonly used models for VAR?

**Suitability:**

3

---

### Meta-Review · Area_Chair_qETX · 2024-06-29

**Recommendation:** Accept (Poster)
**Confidence:** 4

**Metareview:**

The reviewers have major concerns on the motivations, novelty, experimental results and paper writing. After rebuttal, two reviewers increase the score from borderline reject to borderline accept. In summary, three reviewer agree to borderline accept the paper and one reviewer remains borderline reject.

Though there are remaining issues in this paper, the paper has some merits to be accepted into ACM MM.